# Non-Dimensional Star-Identification [note 1]

**DOI:** 10.3390/s20092697

**Published:** 2020-05-09

**Authors:** Carl Leake, David Arnas, Daniele Mortari

**Affiliations:** 1Aerospace Engineering, Texas A&M University, College Station, TX 77843, USA; mortari@tamu.edu; 2Department of Aeronautics and Astronautics, Massachusetts Institute of Technology, Cambridge, MA 02139, USA; arnas@mit.edu

**Keywords:** star identification, non-dimensional, Pyramid, *n*-dimensional *k*-vector

## Abstract

This study introduces a new “Non-Dimensional” star identification algorithm to reliably identify the stars observed by a wide field-of-view star tracker when the focal length and optical axis offset values are known with poor accuracy. This algorithm is particularly suited to complement nominal lost-in-space algorithms, which may identify stars incorrectly when the focal length and/or optical axis offset deviate from their nominal operational ranges. These deviations may be caused, for example, by launch vibrations or thermal variations in orbit. The algorithm performance is compared in terms of accuracy, speed, and robustness to the Pyramid algorithm. These comparisons highlight the clear advantages that a combined approach of these methodologies provides.

## 1. Introduction

If a star tracker is working as intended, a nominal lost-in-space star identification (Star-ID) algorithm can be used to identify stars using only the observed directions of the unknown stars and the on-board star catalog. These algorithms are paramount for determining the attitude of a spacecraft using a star tracker. In terms of speed and robustness, the state-of-the-art algorithm to identify stars in the nominal lost-in-space scenario is the nominal lost-in-space algorithm Pyramid [1]. The Pyramid algorithm recognizes observed stars using the invariance of the angles between observed and cataloged stars. The *k*-vector range searching technique [2,3] is the internal engine of Pyramid that facilitates a quick and robust Star-ID. The Pyramid algorithm is summarized in Appendix B.

While nominal lost-in-space algorithms are typically robust to centroiding error and false stars [1,4,5,6,7,8], they require an accurate estimate of the star tracker’s focal length and optical axis (OA) offset. However, during a spacecraft’s lifetime, environmental effects such as temperature and vibrations may perturb the focal length and OA offset of the star tracker. When this happens, the Pyramid algorithm may become unable to identify stars or, potentially worse, identify stars incorrectly. Hence the need for a Star-ID algorithm that is less sensitive to these changes in camera parameters.

To that end, previous algorithms [9,10,11,12,13] have proposed solutions based on the non-dimensional Star-ID problem, i.e., the identification of stars when the focal length or OA offset of the star tracker were perturbed from their nominal values. However, these algorithms present an important limitation: the non-dimensional database used to obtain the Star-ID must be small. The small non-dimensional databases are obtained by either filtering the data or by using a star tracker with a narrow field-of-view (FOV). Reference [11] is an example of filtering the data to obtain a small non-dimensional database. The star catalogue has 1638 stars, but the star triangles formed by these stars are filtered such that the non-dimensional database contains only 1160 star triangles. Reference [9] is an example of using a small FOV star tracker, 8°×8°, to obtain a small non-dimensional database. Reference [9] uses the maximum and minimum angles of planar triangles to identify stars. Therefore, if the FOV of the star tracker becomes too wide, the resultant non-dimensional star catalog becomes so populated that the difference in angles between adjacent entries are smaller than the centroiding accuracy of the camera. As a result, it becomes nearly impossible to identify a unique star triangle, and thus, the algorithm cannot identify any stars.

This article proposes the non-dimensional Star-ID algorithm (NDSIA), a non-dimensional algorithm that is able to handle much larger star catalogs efficiently. This is accomplished by using the dihedral angles of spherical star triangles rather than the angles of planar triangles, and by considering a database composed of all three spherical triangle angles rather than just one or two of the planar triangles’ angles. The term planar triangles refers to triangles that lie on a flat plane. These triangles have familiar properties, such as the sum of their angles is 180° and the sum of any two sides of the triangle is greater than the third. For readers unfamiliar with spherical geometry, a diagram depicting a spherical triangle is shown in Figure A1 of Appendix A. These two changes prevent the non-dimensional catalog from becoming too large; the problem that plagued previous works. As a consequence of these modifications, the NDSIA must perform orthogonal range searches in a three-dimensional database, as opposed to previous algorithms that only dealt with one- or two-dimensional databases. In general, searching in multidimensional databases is significantly more time-consuming due to the increase in the amount of data, and the lack of a clear ordering methodology for multidimensional elements (i.e., in one dimension it is clear how to sort numerical elements so they can be extracted easily, for example in ascending order, but in *n*-dimensions such a sorting process is more convoluted). Fortunately, recent advances in the *n*-dimensional *k*-vector [14] make the searching process viable for real-time applications. As an example of that, this work shows that the NDSIA takes milliseconds or tens of milliseconds to run, depending on the perturbations of the star tracker parameters.

The remainder of the article is organized as follows. First, the theory for the NDSIA is introduced. Then, a comparison is made between the NDSIA and Pyramid for eight different scenarios, each with different camera perturbations. These eight scenarios showcase the performance of the NDSIA when subject to focal length and OA offset perturbations.

## 2. The Non-Dimensional Star-ID Algorithm

The NDSIA is designed for scenarios where the focal length, *f*, and/or OA offset, $\left[x_{oa},y_{oa}\right]$, may differ from their nominal values, and is meant to be a backup Star-ID algorithm that is only used when the Pyramid algorithm begins to fail. The NDSIA is not meant to replace Pyramid when the star tracker is working nominally, as it requires more memory and a larger computational effort than Pyramid, and it does not provide any benefits when compared with Pyramid under nominal star tracker conditions. As an added benefit, once enough stars are identified using the NDSIA, the actual focal length and OA offset of the camera can be calculated, as shown in Reference [9]. Once these quantities are obtained, a new star database can be generated so that Pyramid can be used again. A skeleton for implementing Pyramid and the NDSIA synergously is given here, but the finer details are left to future work, as the purpose of this article is to introduce and analyze the NDSIA. Typically, the Star-ID is done by Pyramid, and the TASTE test [15], or similar self-consistency check, is used to self-diagnose unsuccessful Pyramid Star-IDs (i.e., incorrect star matches). Based on user specifications, if Pyramid’s Star-IDs are unsuccessful often enough, then the Star-ID is performed by the NDSIA instead. The NDSIA performs the Star-ID until a sufficient number stars have been identified to recalculate the focal length and OA offset of the camera [9]. Finally, a new database is created based on these parameters, and the Star-ID is once again performed by Pyramid.

The non-dimensional algorithm presented here is reminiscent of the algorithms presented in References [9,10,11,12,13,16]. Nevertheless, the non-dimensional algorithm introduced in this work has three major differences when compared with these previous algorithms:The NDSIA identifies stars using the dihedral angles of spherical star triangles. Previous algorithms used angles of planar star triangles [9,10,11,12], the sides of planar star triangles [13], or asterisms composed of four or five stars [16].The NDSIA uses the *n*-dimensional *k*-vector (NDKV) [14] to search among a three-dimensional database. Previous algorithms search among a one- or two-dimensional database [9,10,11,12,13], using searching techniques such as the one-dimensional *k*-vector [3], or by geometrically hashing asterisms of stars and using a hash-based search algorithm [16].The NDSIA performs a final check that uses the identified stars’ interstellar angles to further the confidence in the Star-ID. This check is non-existent in previous algorithms.

In addition, the database size (5 GB) and CPU time required by the algorithm of Reference [16] are not well suited for star tracker applications. Conversely, the database size and CPU time required by NDSIA and algorithms of References [9,10,11,12,13] are.

In previous works, planar star triangles were used to identify stars, because planar triangles are more insensitive to camera perturbations than the interstellar angles used in Pyramid. The dihedral angles of spherical triangles also have this property. However, the dihedral angles of spherical triangles contain more information than planar triangle angles, because planar triangles are the projections on a plane of spherical triangles. A more mathematically rigorous way of stating this is planar triangle angles are subject to the constraint that the sum of the angles must be equal to 180°. Thus, each planar triangle only contains two independent pieces of information. In contrast, spherical triangles do not have this constraint. In fact, the sum of spherical triangle angles is in the range (180°,540°). A numerical study was performed to identify which positions in an image with a given size and focal length generate the largest sum of the three dihedral angles when projected onto the unit sphere. The numerical study concluded that the maximum sum of the three dihedral angles occurs when the three stars are on the border of the image, but their exact positions depend on the image size and the focal length. In summation, each of the three spherical triangle angles provides an independent piece of information. Hence, more information is available when observing three stars as a spherical triangle rather than as a planar triangle.

Nevertheless, previous works [9,10,11,12,13] were only able to search a one- or two-dimensional database quickly enough for real-time applications. Thus, they were not able to use all the information available from spherical triangles. However, recent advances in the NDKV lifted the one- or two-dimensional database restriction. Consequently, the information from all three dihedral angles of spherical triangles can be used to uniquely identify stars. Leveraging this information allows for more entries in the non-dimensional database than in previous works.

### 2.1. Creating the Non-Dimensional Database

This section presents the database structure used by the NDSIA and shows how it is generated. the NDSIA uses a database, called the non-dimensional database, composed of dihedral angles of spherical star triangles and their corresponding stars. A visual representation of the dihedral angles of spherical star triangles and a detailed description of how they are calculated is shown in Appendix A. To construct the non-dimensional database, the algorithm needs to first identify all the potential star triangles that fit within the FOV of the star tracker. Once these admissible triangles have been identified, their dihedral angles are computed using the equations shown in Appendix A. Each spherical triangle becomes an element in the non-dimensional database, where each element is composed of the dihedral angles, sorted in ascending order, followed by the stars that lie at the vertices associated with the sorted dihedral angles.

As mentioned previously, the NDSIA requires more memory than Pyramid. A comparison of the databases used by each of theses algorithms quantifies this statement. Both databases were created in Python on Ubuntu 18.04 with an Intel(R) Core(TM) i5-2400 CPU at 3.10 GHz and 16.0 GB of RAM. Each database was generated from an original star catalog of 1673 stars. The non-dimensional database took 901 s to generate, and contains 6 rows and 2,762,895 columns. It takes up 99.46 MB of memory. The Pyramid database took 0.175 s to generate, and contains 99,422 rows and 3 columns. It takes up 1.59 MB of memory, almost two orders of magnitude less memory than the NDSIA. While the database generation can be done offline, so the computational time is not necessarily important, it is included here for completeness.

### 2.2. The Non-Dimensional Star-ID Algorithm

The identification process of the NDSIA is similar to that of Pyramid [1], but with two key differences. First, the NDSIA makes use of two reference stars rather than one, and second, the NDSIA uses a final check at the end of the identification process that Pyramid does not. The flowchart in Figure 1 summarizes the NDSIA algorithm. The rectangular boxes in Figure 1 represent processes. Each process contains a set of specific steps that are described in detail in the subsections that follow.

#### 2.2.1. Identify a Unique Star Triangle

A unique star triangle is a star triangle that only returns one match when searched for in the non-dimensional star database. A star triangle is identified by its dihedral angles. Thus, to find a unique star triangle, the dihedral angles must be computed using the equations in Appendix A. The dihedral angles are sorted in ascending order, to match the non-dimensional database, and an orthogonal range is created by adding and subtracting the quantity ϵ from the dihedral angles. Then, the NDKV orthogonal range searching algorithm [14] is used to find all possible triangles in the database that fit this range. The parameter ϵ can be used to tune the performance of the NDSIA. If ϵ is smaller, the Star-ID process will be completed a lower percentage of the time, but the confidence in the Star-ID increases. If the parameter ϵ is larger, the opposite is true. In this work, ϵ is chosen to be three standard deviations of the star tracker’s centroiding error. If the range search only returns one match and the $L_{2}$ norm of the difference between the spherical triangle angles in the camera frame and those in the database is less than ϵ, then that triangle is a unique star triangle. The latter of these conditions can be mathematically expressed as,
(1)$$\sqrt{{\left(A-A_{d}\right)}^{2}+{\left(B-B_{d}\right)}^{2}+{\left(C-C_{d}\right)}^{2}}<\epsilon ,$$
where *A*, *B*, and *C* are the dihedral angles computed in the camera frame and $A_{d}$, $B_{d}$, and $C_{d}$ are the angles of the matching spherical triangle in the non-dimensional database. If the range search returns zero matches, more than one match, or the inequality in Equation (Equation 1) is not satisfied, then that triangle is not a unique star triangle.

All possible combinations of three stars in the star tracker’s FOV are tested until a unique star triangle is identified. The method used to test all three star combinations is known as the star kernel generator. Naturally, one may be inclined to use a star kernel generator that loops through the stars in order to identify a unique star triangle. However, this is not the most efficient method. For example, if the first star in the loop is a false-star (i.e., noise in the camera rather than a true star), then all triangles using that star will have to be tested before moving onto the next star. Hence, a more efficient star kernel generator is employed to ensure that all triangles are tested, but that avoids using one particular star over and over again: the pattern shifting star kernel generator from Reference [17]. Please note that the enhanced pattern shifting star kernel generator from Reference [17] was tested as well, but did not show any noticeable speed improvements over the pattern shifting star kernel generator for the numerical tests of the NDSIA shown in this article.

If no unique star triangle is found, then the NDSIA exits and reports it cannot identify any stars. If a reference star is found, then the algorithm moves onto the next step. Let the candidate stars of the unique star triangle found in this step be denoted as the *i*, *j*, and *k* stars. Let the unique star triangle itself be referred to as the {i,j,k} triangle.

#### 2.2.2. Identify a First Reference Star

To minimize the chance of identifying stars incorrectly, two reference stars are used to verify the *i*, *j*, and *k* stars. Let *r* denote the first reference star. The first reference star is considered identified if and only if the star triangles {r,i,j}, {r,i,k}, and {r,j,k} are unique star triangles. All stars that are not the *i*, *j*, and *k* stars are tested as potential reference stars until one is found. If a reference star is found, then the algorithm continues on to the next step. Otherwise, the algorithm moves back a step and continues trying to find a different set of three stars to use as the {i,j,k} triangle.

#### 2.2.3. Identify a Second Reference Star

To minimize the chance of identifying stars incorrectly, a second reference star is used to verify the *i*, *j*, *k*, and *r* stars. Let $r_{2}$ denote the second reference star. The second reference star is considered identified if and only if the star triangles $\left\{i,j,r_{2}\right\}$, $\left\{i,k,r_{2}\right\}$, $\left\{j,k,r_{2}\right\}$, $\left\{i,r,r_{2}\right\}$, $\left\{j,r,r_{2}\right\}$, and $\left\{k,r,r_{2}\right\}$, are unique star triangles. All stars that are not the *i*, *j*, *k*, and *r* stars are tested as potential second reference stars until one is found. If a second reference star is found, then all five stars, *i*, *j*, *k*, *r*, and $r_{2}$, are considered identified and are recorded. Otherwise, the algorithm moves back a step and continues trying to find a different first reference star *r*.

#### 2.2.4. Identify Remaining Stars

After identifying the *i*, *j*, *k*, *r*, and $r_{2}$ stars, the remaining stars in the frame are identified using the same process as the first reference star. That is, any remaining star *s* can be identified if and only if the star triangles {s,i,j}, {s,i,k}, and {s,j,k} are unique star triangles. If the aforementioned triangles are not unique star triangles, then star *s* is discarded.

#### 2.2.5. Final Check Using Interstellar Angles

Once all the stars that can be identified have been identified, a final check is performed using the identified stars’ interstellar angles. This is done by calculating the interstellar angles between every possible star pair from the subset of stars already identified. Appendix A shows how to calculate the interstellar angle between two stars. If any of the interstellar angles obtained is greater than the FOV of the camera, then an error has been made in the Star-ID process. If this happens, then no stars can be accurately identified, and thus, the algorithm reports that it cannot identify any stars.

## 3. The Non-Dimensional Star-ID Algorithm Compared with Pyramid

In this section, the NDSIA is compared with Pyramid using the following five metrics:Number of scenes wherein the Star-ID was completed. This metric shows whether the algorithm attempted to perform a Star-ID or not. This metric is reported as a percentage of the total number of scenes in the test.Number of successful Star-IDs. This metric is calculated as the ratio of the number of times that all stars identified by the algorithm were identified correctly, to the number of times the Star-ID was completed. This ratio is reported as a percentage.Average time that it takes to perform one Star-ID. This metric is reported in milliseconds.Number of scenes wherein Pyramid did not complete the Star-ID and the NDSIA completed the Star-ID successfully. This metric is reported as a percentage of the number scenes wherein Pyramid did not complete the Star-ID.Number of scenes wherein Pyramid completed the Star-ID unsuccessfully (i.e., at least one star was identified incorrectly) and the NDSIA completed the Star-ID successfully. This metric is reported as a percentage of the number of scenes wherein Pyramid completed the Star-ID unsuccessfully.

These five metrics are used to evaluate the NDSIA’s performance over a set of eight tests designed to cover different star tracker situations. Table 1 describes the parameters used for each test.

### 3.1. Test Conditions

Each test contained 1000 randomly oriented scenes, where a scene consists of performing the Star-ID and estimating the attitude. The random orientations were generated via QR decomposition of a randomly generated 3×3 matrix. The seed used to generate the random 3×3 matrix was fixed such that all tests used the same random orientations. All of the tests included in this section were performed in C++ on a computer running Ubuntu 18.04 with an Intel(R) Core(TM) i5-2400 CPU at 3.10 GHz and 16.0 GB of RAM. All run times were calculated using the system_clock function in the C++ boost chrono library. The parameters of the virtual star tracker used in these tests are shown in Table 2, where U[0,5] represents the uniform distribution of integers in the range [0,5] and σ denotes the standard deviation of a normal distribution.

The centroiding error was applied to the star measurements in the camera frame using the methodology shown in Equation (Equation 2).
(2)$$\begin{array}{cc} \theta & ∼N(0,\sigma^{2})\\ e & =v\times {\hat{b}}_{t}\\ \hat{e} & =e/||e||\\ C_{p} & =C_{p}\left(\hat{e},\theta \right)\\ {\hat{b}}_{e} & =C_{p}\;{\hat{b}}_{t} \end{array}$$
where $N(\mu ,\sigma^{2})$ is the normal distribution with mean μ and variance $\sigma^{2}$, σ is the centroid error of the camera, ${\hat{b}}_{t}\in R^{3}$ is a unit vector that points in the true direction of the star in the camera reference frame, ${\hat{b}}_{e}\in R^{3}$ is the observed unit vector, affected by centroid error, $v\in R^{3}$ is a random vector, and the function $C_{p}\left(\hat{e},\theta \right)$ produces an attitude matrix with principle axis $\hat{e}$ and principle angle θ. The function used to calculate $C_{p}$ given $\hat{e}$ and θ is,
$$C_{p}\left(\hat{e},\theta \right)=I_{3\times 3}\;\cos\theta +\left(1-\cos\theta \right)\;\hat{e}{\hat{e}}^{T}-\left[\hat{e}\times \right]\;\sin\theta$$
where $I_{3\times 3}$ is the 3×3 identity matrix and $\left[\hat{e}\times \right]$ is the skew-symmetric matrix formed using the components of $\hat{e}$.

The focal length and OA offset perturbations are applied to the star measurements in the camera frame using the methodology shown in Equation (Equation 3),
(3)$$\begin{array}{cc} \left\{\begin{array}{c} x_{c}\\ y_{c} \end{array}\right\} & =\left[\begin{array}{ccc} -f-\delta f & 0 & 0\\ 0 & -f-\delta f & 0 \end{array}\right]\frac{{\hat{b}}_{e}}{{\hat{b}}_{e}\left(3\right)}\\ b_{p} & =\left\{\begin{array}{c} \pm x_{c}-x_{oa}+\delta x\\ \pm y_{c}-y_{oa}+\delta x\\ f+\delta f \end{array}\right\}\\ {\hat{b}}_{p} & =\frac{b_{p}}{|\;b_{p}\;|}, \end{array}$$
where ${\hat{b}}_{e}\left(3\right)$ is the third component of the vector ${\hat{b}}_{e}$, δf is the focal length perturbation, δx and δy are the OA offset perturbations, and ${\hat{b}}_{p}$ is the unit vector pointing to the star whose centroid coordinates are $\left[x_{c},y_{c}\right]$ on the imager. The “±” sign appearing in Equation (Equation 3) depends on the imager *x* and *y* axes directions. Please note that Equation (Equation 3) assumes that the star tracker is modelled as an ideal pin-hole camera.

### 3.2. Results

Table 3 gives a summary of the performance and results of Pyramid and the NDSIA on the eight tests. For each algorithm, three columns are included that contain the first three metrics described earlier. The first column, $n_{id}\left(\%\right)$, gives the percentage of scenes where the Star-ID was completed, and the second column, $n_{+id}$, shows the percentage of completed Star-ID runs with a successful Star-ID (i.e., a Star-ID wherein all stars identified by the algorithm are identified correctly). The third column shows the average time each algorithm took to perform the Star-ID in milliseconds. In the joint statistics column of the table, the remaining two metrics are reported.

In general, Table 3 shows that the NDSIA is reliable, even when the camera parameters are perturbed, as it never completes a Star-ID unsuccessfully (i.e., it never incorrectly identifies a star). Furthermore, in every test, the NDSIA is able to successfully complete the Star-ID in more than 50% of the scenes wherein Pyramid completes the Star-ID unsuccessfully. The joint statistics for the NDSIA completing Star-IDs successfully where Pyramid does not complete the Star-ID at all are less impressive, but still show that the NDSIA maintains some benefit in this regard over Pyramid in all tests with focal length perturbations. Finally, Table 3 shows that the NDSIA takes on the order of tens of milliseconds or less to perform the Star-ID, a speed that is suitable for many real-time applications.

Test 1 shows that when the star tracker is working nominally Pyramid and the NDSIA are each able to identify the stars correctly, with one exception; in one case, Pyramid performs the Star-ID incorrectly. However, in this case, the final attitude error, 0.011 degrees, is still small. Pyramid is able to identify stars more often than the NDSIA, and Pyramid is more than an order of magnitude faster than the NDSIA. This is expected, as the NDSIA is meant to be used only when the Pyramid algorithm begins to fail.

Test 2 and Test 3 are the first two examples of such situations. These tests were performed with focal length perturbations. Compared to the nominal case, the Pyramid algorithm performance degrades significantly. The most alarming change in the performance of the Pyramid algorithm is the reduction in the percentage of scenes with a successful Star-ID. In Test 2 and Test 3, the Star-ID performed by Pyramid can no longer be trusted. In contrast, the percentage of the NDSIA scenes with a successful Star-ID remains unchanged from the nominal case. However, the number of Star-IDs that can be completed is reduced significantly when compared with the nominal case. In addition, the time required to complete a Star-ID increases due to the focal length perturbations.

In Test 4 and Test 5, the performance of the Pyramid algorithm remains almost unchanged when compared with Test 1, the nominal case. The reason is small OA offset perturbations have little effect on the interstellar angles between stars for most locations on the imager. Therefore, the Pyramid algorithm is still able to perform the Star-ID. Conversely, the small OA offset perturbations have a larger effect on the dihedral angles between the stars. Thus, the number of times that the Star-ID can be completed by the NDSIA is reduced as the OA offset increases. However, the percentage of scenes with a successful Star-ID remains unchanged from the nominal case. Therefore, while the NDSIA may not be able to perform the Star-ID process as often as Pyramid when there is only an OA offset perturbation, the Star-ID provided by the NDSIA can still be trusted.

Test 6 and Test 7 show the performance of the two algorithms when subject to both focal length perturbations and OA offset perturbations. In these two tests, there is again a significant degradation in the performance of the Pyramid algorithm, and the number of Star-IDs that both algorithms can complete is reduced when compared with Test 1, the nominal case.

Test 8 shows that the NDSIA is robust to changes in the centroiding accuracy of the camera, because the performance of the NDSIA in Test 8 is similar to the performance of the NDSIA in Test 6—Test 6 is the same as Test 8 but with a lower centroiding error. Comparing Pyramid and the NDSIA on Test 8 reveals that the NDSIA does better in terms of percentage of Star-IDs completed and percentage of Star-IDs completed successfully. The Pyramid algorithm is slightly faster than the NDSIA in Test 8. Comparing the Pyramid results for Test 8 with the Pyramid results for Test 6 shows that Pyramid is able to perform more Star-IDs in Test 8 than Test 6. The larger centroiding error absorbs some of the error due to the focal length and OA offset perturbations. In other words, in many scenes, the position error of stars in the camera frame due to focal length and OA offset perturbations is less than the typical centroiding error in Test 8. Thus, with the increased centroiding error, and therefore an increased range when performing the associated range searches, Pyramid is able to identify stars in more scenes.

Based on the results of Test 8, one is naturally led to wonder if the performance of Pyramid can be improved by increasing the range it uses when performing range searches in its database. Please note that in Test 8 both the actual camera centroiding error and the range used in Pyramid were modified. In contrast, here, only the range Pyramid uses will be modified, the actual camera centroiding error will be held constant, using the value given in Table 2. Table 4 shows how the performance of Pyramid is impacted by changing the size of the range search. The left most column in Table 4 gives the standard deviation, σ, that Pyramid assumes the star tracker camera has; hence, the range that Pyramid uses when searching the database of interstellar angles is ±3σ. The focal length and OA offset perturbations used to create Table 4 are the same as those used in Test 6. Hence, the first row of Table 4 is identical to that of Test 6. Each row in Table 4 was created using 1000 scenes, the same as in the previous eight tests.

Table 4 shows that as the range that Pyramid uses increases, so does the percentage of tests with completed Star-IDs. Moreover, the percentage of successful star IDs remains approximately constant. As a result, the total number of tests for which Pyramid completes a successful Star-ID increases as the range increases. However, this also means that there are still a notable number of cases for which Pyramid identifies stars incorrectly. Hence, even by varying the range that Pyramid uses to search in the database, the Pyramid algorithm does not outperform the NDSIA.

Histograms of attitude error are provided for each test to quantify how much the perturbations ultimately affect the attitude estimation. The histograms only include the attitude estimation error for scenes where the Star-ID was completed successfully, as almost all scenes with an incorrect Star-ID have a large attitude error. The q-method is used to estimate the attitude for each algorithm [18]. Therefore, the only variable that affects the attitude estimation error is the number of stars identified by each algorithm in each scene. Figure 2 shows the histograms of attitude error for Pyramid and the NDSIA for Test 1. This figure shows that the two algorithms have similar attitude error distributions, but that Pyramid completed more Star-IDs than the NDSIA.

Figure 3 and Figure 4 show histograms of the attitude error for Pyramid and the NDSIA for Test 2 and Test 3. These histograms clearly show that the NDSIA is more accurate on average than the Pyramid algorithm. Moreover, comparing Test 2 and Test 3 to Test 1 reveals that the attitude error of the NDSIA increases slightly as the focal length perturbation increases.

Figure 5 and Figure 6 show the attitude error histograms for Pyramid and the NDSIA for Test 4 and Test 5. These figures show that the attitude error distributions for the two algorithms are similar for both tests; however, in each case, the NDSIA completed fewer Star-IDs than the Pyramid algorithm. When compared to Test 1, the attitude error of both algorithms increases with the increase in the OA offset perturbation.

Figure 7 and Figure 8 show the attitude error histograms for Pyramid and the NDSIA for Test 6 and Test 7. These figures show that on average the NDSIA is more accurate than Pyramid. In these tests, the NDSIA is able to complete the Star-ID process in more cases than Pyramid.

Figure 9 shows the attitude error histograms for Pyramid and the NDSIA for Test 8. This figure shows that on average the NDSIA is more accurate than Pyramid. In Test 8, the NDSIA was able to complete the Star-ID process more times than Pyramid. Comparing the results of this test with Test 6 reveals that the performance of the NDSIA is similar between the two tests, whereas Pyramid’s performance improves noticeably in Test 8.

## 4. Future Work

Although an outline for implementing the NDSIA and Pyramid in tandem is given in this article, future research should work out the finer details of said implementation. Furthermore, future research should investigate how the myriad star kernel generators, shifting necklace [17] and greedy scene elimination [19] to name a few, impact the performance of the NDSIA. In addition, the NDSIA considers the magnitudes of the dihedral angles that make up the spherical triangle, but not their orientation. Calculating the orientation of the triangles as is done Reference [11] would add another differentiating parameter, and thus another dimension, to the non-dimensional database that could be used to further the confidence of the Star-ID. Moreover, increasing the dimensionality of the non-dimensional database by using three dihedral angles rather than one or two planar angles allowed the NDSIA’s non-dimensional database to have a larger number of elements compared to previous algorithms. Thus, adding a fourth dimension would allow for an even larger non-dimensional database. While adding another dimension increases the difficulty of the orthogonal range search, the *n*-dimensional *k*-vector has been shown to handle four-dimensional range searches with ease [14].

## 5. Conclusions

This work introduces the Non-Dimensional Star-Identification Algorithm (NDSIA), a real-time star identification algorithm for star trackers on board spacecraft. The NDSIA is devised to complement other nominal lost-in-space algorithms, for instance Pyramid. When the nominal lost-in-space algorithm fails due to perturbations of the focal length and/or the OA offset in the star tracker, the NDSIA is still able to identify stars. Moreover, not only can the NDSIA be used to identify stars when the nominal lost-in-space algorithm fails, it can be used to re-calibrate the star tracker.

The idea behind the NDSIA is to identify stars via a database containing information about spherical star triangles that can be generated from a set of stars in a star catalogue. In particular, each element from this database contains both the stars that generated the star triangle, and values of the dihedral angles that these stars form in a unit 3D sphere. Each time that an identification is required, three candidate stars from the frame are selected and checked against the database. If a unique match is found, then two additional stars, called reference stars, from the frame are selected and used to generate all nine possible star triangles between each reference star and the stars in the original triangle. If these nine star triangles also produce unique matches in the database, then the algorithm identifies this subset of five stars. The original three stars are used to continue with the identification of the remaining candidate stars from the image. Each remaining star is identified using a process similar to the first reference star. Finally, once the process is finished, the interstellar angles between each pair of identified stars is used as a final check to ensure the star identification was performed correctly.

It is important to note that in order to perform the identification of a star triangle in the algorithm database, a three-dimensional search is required, where each dimension of the search corresponds to each one of the dihedral angles included in the database. Using a three-dimensional database significantly reduces the number of star triangle combinations for a given dihedral angle tolerance at the cost of making the searching process more difficult. Nevertheless, the increased search difficulty is solved in the NDSIA by using the *n*-dimensional *k*-vector (NDKV) algorithm.

Through a series of eight tests in this article, the NDSIA was demonstrated to be robust to changes in both the focal length and OA offset of a star tracker; it provided a reliable Star-ID even when these camera parameters were perturbed. In contrast, algorithms such as Pyramid showed an alarming increase of failures or incorrect Star-IDs when subject to focal length perturbations. Under perturbations just in the OA offset, the tests performed in this work showed little effect on Pyramid.

One important property of the NDSIA is that the algorithm can be used to compute the new values of the focal length and/or OA offset of the star tracker camera. Using this information, it is possible to update the star database on board the spacecraft to take into account these perturbations. That way, the nominal star identification algorithm, which usually has a better speed performance and lower computational cost, can be used again as it is more efficient.

The results presented in this work show that using a combination of the NDSIA and Pyramid algorithms provides a fast and robust methodology for dealing with the real-time star identification problem even under the effect of perturbations in the focal length and/or the OA offset of star trackers. Pyramid is to be used whenever the star tracker is functioning nominally, and the NDSIA when perturbations occur. Together, the strengths of one algorithm complement the weaknesses of the other, increasing the reliability of the system as a whole.

## Figures and Tables

**Figure 1 sensors-20-02697-f001:**
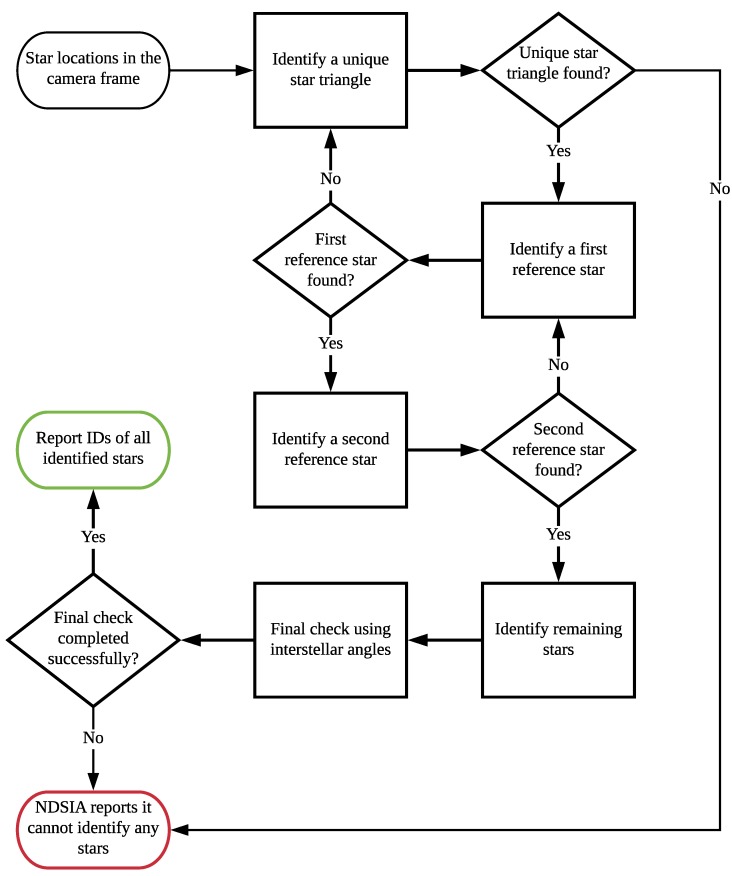
The NDSIA Flowchart.

**Figure 2 sensors-20-02697-f002:**
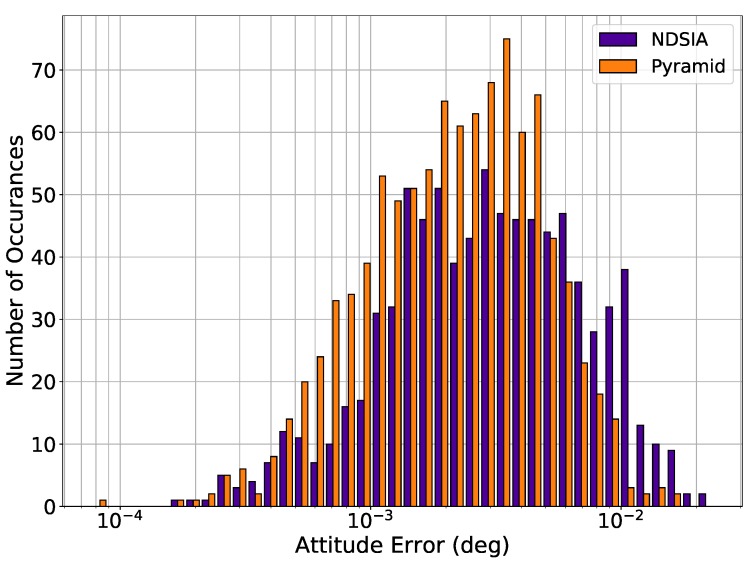
Test 1, Nominal Test, Attitude Estimation Error for Scenes with a Successful Star-ID.

**Figure 3 sensors-20-02697-f003:**
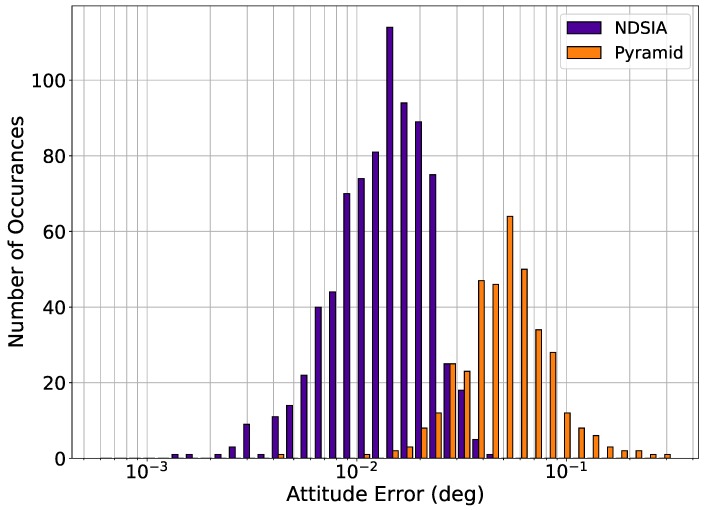
Test 2, Small Focal Length Perturbation, Attitude Estimation Error for Scenes with a Successful Star-ID.

**Figure 4 sensors-20-02697-f004:**
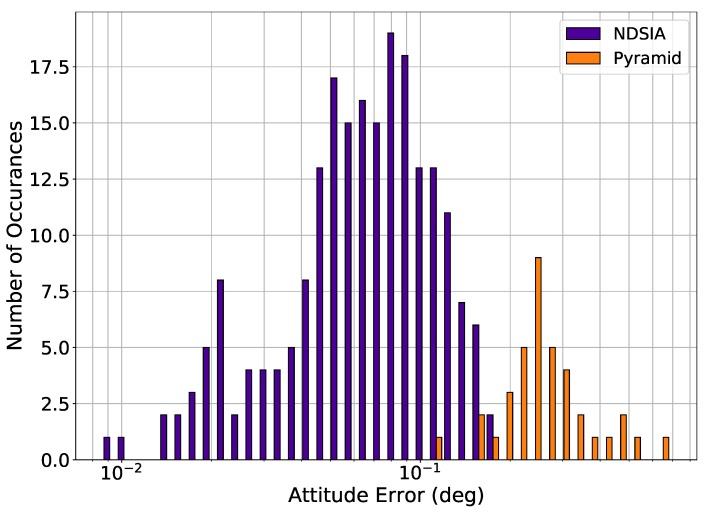
Test 3, Large Focal Length Perturbation, Attitude Estimation Error for Scenes with a Successful Star-ID.

**Figure 5 sensors-20-02697-f005:**
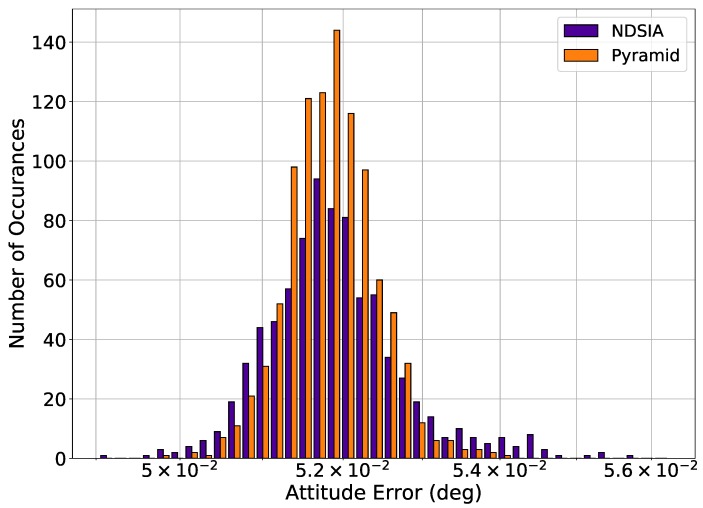
Test 4, Small OA Offset Perturbation, Attitude Estimation Error for Scenes with a Successful Star-ID.

**Figure 6 sensors-20-02697-f006:**
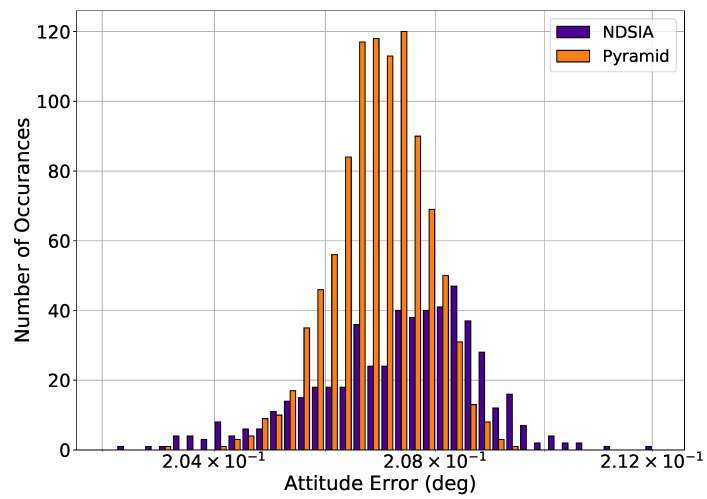
Test 5, Large OA Offset Perturbation, Attitude Estimation Error for Scenes with a Successful Star-ID.

**Figure 7 sensors-20-02697-f007:**
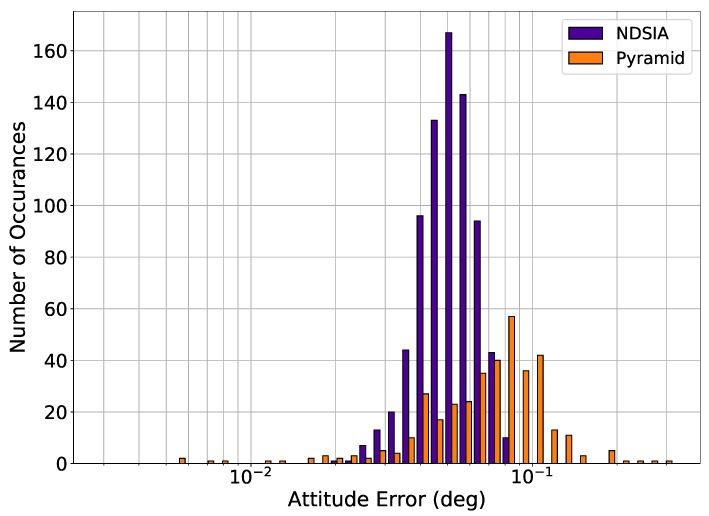
Test 6, Small Focal Length and OA Offset Perturbation, Attitude Estimation Error for Scenes with a Successful Star-ID.

**Figure 8 sensors-20-02697-f008:**
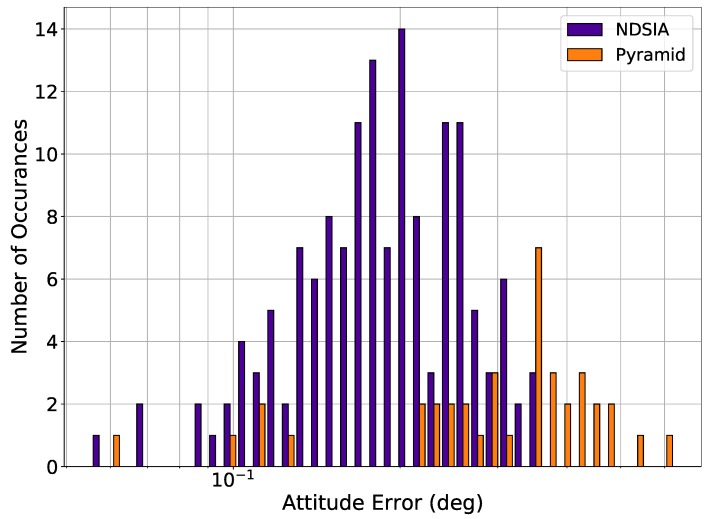
Test 7, Large Focal Length and OA Offset Perturbation, Attitude Estimation Error for Scenes with a Successful Star-ID.

**Figure 9 sensors-20-02697-f009:**
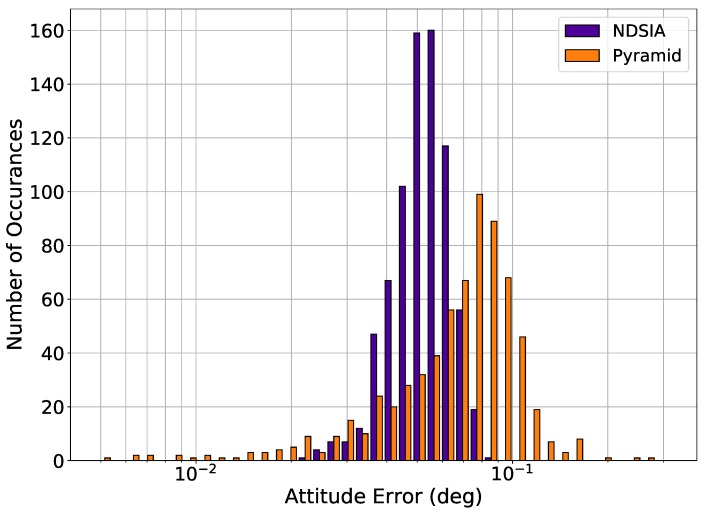
Test 8, Small Focal Length and OA Offset Perturbation and Larger Centroiding Error, Attitude Estimation Error for Scenes with a Successful Star-ID.

**Table 1 sensors-20-02697-t001:** Test descriptions.

Test Number	Focal Length Perturbation (% of Nominal Focal Length)	Optical Axis Offset Perturbation (% of Half the Imager Width)	Camera Centroiding Error Standard Deviation (Arcsec)
1	0	0	10
2	0.5	0	10
3	2.0	0	10
4	0	0.5	10
5	0	2.0	10
6	0.5	0.5	10
7	2.0	2.0	10
8	0.5	0.5	15

**Table 2 sensors-20-02697-t002:** Star tracker parameters.

Virtual Star Tracker Parameter	Value
1σ centroid error	10 arcseconds
Star magnitude threshold	5.0
Number of false stars	U[0,5]
Number of pixel rows	1024
Number of pixel columns	1024
Pixel pitch	0.018 mm
Focal length	50.47 mm
Number of stars in star catalogue	1673
Number of interstellar angles in Pyramid database	99,422
Number of star triangles in non-dimensional database	2,762,895

**Table 3 sensors-20-02697-t003:** Comparison between the Pyramid algorithm and the NDSIA.

Test	Pyramid			NDSIA			Joint Statistics	
	$n_{\mathrm{id}}$ (%)	$n_{+\mathrm{id}}$ (%)	$t_{\mathrm{avg}}$ (ms)	$n_{\mathrm{id}}$(%)	$n_{+\mathrm{id}}$ (%)	$t_{\mathrm{avg}}$ (ms)	Incomplete Pyramid Star-ID Scenes Completed Successfully by the NDSIA (%)	Unsuccessful Pyramid Star-ID Scenes Completed Successfully by the NDSIA (%)
1	100	99.9	0.139	84.2	100	2.81	N/A	100
2	41.8	90.7	10.8	79.4	100	3.80	68.0	92.3
3	9.20	41.3	27.8	21.5	100	24.3	17.8	59.3
4	100	99.9	0.141	81.8	100	3.40	N/A	100
5	100	99.9	0.140	53.5	100	12.6	N/A	100
6	41.7	89.9	10.9	77.2	100	4.32	64.5	95.2
7	10.2	38.2	27.6	14.7	100	29.4	10.6	54.0
8	75.5	90.3	6.92	75.9	100	7.06	46.5	91.8

**Table 4 sensors-20-02697-t004:** Pyramid performance as a function of range search size.

σ Used for Pyramid Range Search (Arcsec)	$n_{\mathrm{id}}$ (%)	$n_{+\mathrm{id}}$ (%)	$t_{\mathrm{avg}}$ (ms)
10	41.7	89.9	10.9
15	74.6	90.5	7.16
20	93.3	90.7	3.63
25	98.2	88.8	2.65
30	99.3	90.0	2.93
35	99.8	89.4	4.05
40	99.9	89.8	7.26

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
