# Peer review of "Non-Dimensional Star-Identification†"

_sensors, 2020, doi:10.3390/s20092697_

Round 1

Reviewer 1 Report

The manuscript presents a novel algorithm for star tracker attitude determination in lost-in-space mode. It is tailored to be robust with respect to uncertainties of the parameters of the optical system. The algorithm is well described as well as its performances which are compared to those of an efficient standard algorithm by the same authors. The present manuscript is the upgraded version of a conference paper.

Very limited modifications are suggested.

In the introduction, the detailed description of the “reference” standard “Pyramid” seems to be excessively detailed (even non needed at all), at least badly placed in the manuscript.

All over the manuscript, description of the algorithm step are often repeated. As an example, in section 2.2 and subsections form 2.2.1 to 2.2.4, many issues are repeated. The authors should synthesize and eliminate duplicate descriptions.

The reference section is definitely “polarized”: seven out of ten items are by one of the authors of the manuscript… This is not adequate for a journal paper; the section shall be revised by widening the variety of sources in particular for the scenario description in the Introduction section.

Check the Abbreviations list. At list one abbreviation in use (NDVK) is missing.

Round 2

Reviewer 2 Report

To follow up to previous suggestion about reducing confusion between acronyms, perhaps ND-LISA and P-LISA (for non-dimensional and pyramid, respectively).
